# RETHINKING BACKDOOR DATA POISONING ATTACKS IN THE CONTEXT OF SEMI-SUPERVISED LEARNING

## ABSTRACT

Semi-supervised learning methods can train high-accuracy machine learning models with a fraction of the labeled training samples required for traditional supervised learning. Such methods do not typically involve close review of the unlabeled training samples, making them tempting targets for data poisoning attacks. In this paper we investigate the vulnerabilities of semi-supervised learning methods to backdoor data poisoning attacks on the unlabeled samples. We show that a simple poisoning attack that influences the distribution of the poisoned samples' predicted labels is highly effective - achieving an average attack success rate of 93.6%. We introduce a generalized attack framework targeting semi-supervised learning methods to better understand and exploit their limitations and to motivate future defense strategies.

## 1 INTRODUCTION

Machine learning models have achieved high classification accuracy through the use of large, labeled datasets. However, the creation of diverse datasets with supervised labels is time-consuming and costly. In recent years, semi-supervised learning methods have been introduced which train models using a small set of labeled data and a large set of unlabeled data. These models achieve comparable classification accuracy to supervised learning methods while reducing the necessity of human-based labeling. The lack of a detailed human review of training data increases the potential for attacks on the training data.

Data poisoning attacks adversarially manipulate a small number of training samples in order to shape the performance of the trained network at inference time. Backdoor attacks, one type of data poisoning attack, introduce a backdoor (or an alternative classification pathway) into a trained model that can cause sample misclassification through the introduction of a trigger (a visual feature that is added to a poisoned sample) (Gu et al., 2017). We focus our analysis on backdoor attacks which poison the unlabeled data in semi-supervised learning. In this setting, backdoors must be introduced in the absence of training labels associated with the poisoned images. Recent semi-supervised learning methods achieve high accuracy with very few labeled samples (Xie et al., 2020; Berthelot et al., 2020; Sohn et al., 2020) using the strategies of pseudolabeling and consistency regularization which introduce new considerations when assessing the risk posed by backdoor attacks. Pseudolabeling assigns hard labels to unlabeled samples based on model predictions (Lee et al., 2013) and is responsible for estimating the training labels of unlabeled poisoned samples. Consistency regularization encourages augmented versions of the same sample to have the same network output (Sajjadi et al., 2016) and requires attacks to be robust to significant augmentations.

In this paper we analyze the impact of backdoor data poisoning attacks on semi-supervised learning methods by first reframing the attacks in a setting where pseudolabels are used in lieu of training labels and then highlighting a vulnerability of these methods to attacks which influence expected pseudolabel outputs. We identify characteristics of successful attacks, evaluate how those characteristics can be used to more precisely target semi-supervised learning, and use our insights to suggest new defense strategies. We make the following contributions:

- We show simple, black-box backdoor attacks using adversarially perturbed samples are highly effective against semi-supervised learning methods, emphasizing the sensitivity of attack performance to the pseudolabel distribution of poisoned samples.

- We analyze unique dynamics of data poisoning during semi-supervised training and identify characteristics of attacks that are important for attack success.
- We introduce a generalized attack framework targeting semi-supervised learning.

## 2 BACKGROUND

### 2.1 DATA POISONING

We focus on integrity attacks in data poisoning which maintain high classification accuracy while encouraging targeted misclassification. Instance-targeted attacks and backdoor attacks are two types of integrity attacks. Instance-targeted attacks aim to cause a misclassification of a specific example at test time (Shafahi et al., 2018; Zhu et al., 2019; Geiping et al., 2020; Huang et al., 2020; Aghakhani et al., 2021). While an interesting and fruitful area of research, we do not consider instance-targeted attacks in this paper and instead focus on backdoor attacks. Traditional backdoor attacks introduce triggers into poisoned images during training and adapt the images and/or the training labels to encourage the network to ignore the image content of poisoned images and only focus on the trigger (Gu et al., 2017; Turner et al., 2018; Saha et al., 2020; Zhao et al., 2020). They associate the trigger with a specific target label $y_t$.

There are two types of backdoor data poisoning attacks against supervised learning which use different strategies to encourage the creation of a backdoor: dirty label attacks which change the training labels from the ground truth label (Gu et al., 2017) and clean label attacks which maintain the ground truth training label while perturbing the training sample in order to increase the difficulty of sample classification using only image-based features (Turner et al., 2019; Saha et al., 2020; Zhao et al., 2020). In both of these attacks, the labels are used to firmly fix the desired network output even as the images appear confusing due to perturbations or having a different ground truth class. Greater confusion encourages the network to rely on the triggers, a constant feature in the poisoned samples.

### 2.2 SEMI-SUPERVISED LEARNING

The goal of semi-supervised learning is to utilize unlabeled data to achieve high accuracy models with few labeled samples. This has been a rich research area with a variety of proposed techniques (Van Engelen & Hoos, 2020; Yang et al., 2021). We focus on a subset of recent semi-supervised learning techniques that have significantly improved classification performance (Xie et al., 2020; Berthelot et al., 2020; Sohn et al., 2020). These techniques make use of two popular strategies: consistency regularization and pseudolabeling. Consistency regularization is motivated by the manifold assumption that transformed versions of inputs should not change their class identity. In practice, techniques that employ consistency regularization encourage similar network outputs for augmented inputs (Sajjadi et al., 2016; Miyato et al., 2018; Xie et al., 2020) and often use strong augmentations that significantly change the appearance of inputs. Pseudolabeling uses model predictions to estimate training labels for unlabeled samples (Lee et al., 2013).

### 2.3 DATA POISONING IN SEMI-SUPERVISED LEARNING

While the focus of data poisoning work to date has been on supervised learning, there is recent work focused on the impact of data poisoning attacks on semi-supervised learning. Poisoning attacks on labeled samples have been developed which target graph-based semi-supervised learning methods by focusing on poisoning labeled samples that have the greatest influence on the inferred labels of unlabeled samples (Liu et al., 2019a; Franci et al., 2022). Carlini (2021) introduced a poisoning attack on the unlabeled samples which exploits the pseudolabeling mechanism. This is an instance-targeted attack which aims to propagate the target label from confident target class samples to the target samples (from a non-target class) using interpolated samples between them. Feng et al. (2022) poisons unlabeled samples using a network that transform samples so they appear to the user's network like the target class. Unlike the the traditional goal of backdoor attacks of introducing a backdoor associated with static triggers, they aim to adapt the decision boundary to be susceptible to future transformed samples.

Yan et al. (2021) investigate perturbation-based attacks on unlabeled samples in semi-supervised learning similar to us, but find a simple perturbation-based attack has low attack success. Rather they

suggest an attack (called DeHiB) that utilizes a combination of targeted adversarial perturbations and contrastive data poisoning to achieve high attack success. We show settings in which simple perturbation-based attacks are highly successful. Additionally, in Section 5.1, we discuss how our generalized attack framework encompasses the targeted adversarial perturbations used in DeHiB.

# 3 BACKDOOR ATTACKS IN THE CONTEXT OF SEMI-SUPERVISED LEARNING

## 3.1 ATTACK THREAT MODEL

We consider a setting in which a user has a small amount of labeled data $\mathcal{X}$ for training a classification model. This limited labeled data is not enough to achieve the user's desired classification accuracy, so they collect a large amount of unlabeled data $\mathcal{U}$ from less trusted sources and train their model using the FixMatch semi-supervised learning method (Sohn et al., 2020) to improve accuracy. The adversary introduces poisoned samples $\mathcal{U}_p$ into the unlabeled dataset with the goal of creating a strong backdoor in the trained network, resulting in samples being classified as a chosen target class $y_t$ when a trigger is present. To evade detection, the adversary tries to introduce this backdoor as soon as possible in training and maintain a high classification accuracy in the model trained with the poisoned samples. Because the poisoned samples are only included in the unlabeled portion of the training data, the adversary can only control the image content for the poisoned samples and not the training labels. The adversary does not have access to the user's network architecture.

## 3.2 FIXMATCH DETAILS

FixMatch achieves high classification accuracy with very few labeled samples. It is important to understand details of FixMatch (and similar methods) when aiming to evaluate its potential vulnerability to backdoor attacks. During training, the user has $N_\ell$ labeled samples $\mathcal{X} = \{\boldsymbol{x_i} : i \in (1, ..., N_\ell)\}$ and $N_u$ unlabeled samples $\mathcal{U} = \{\boldsymbol{u_i} : i \in (1, ..., N_u)\}$. The supervised loss term is the standard cross-entropy loss on the labeled samples. The unique characteristics of FixMatch are incorporated in the unsupervised loss term which utilizes pseudolabeling and consistency regularization. FixMatch approximates supervised learning by estimating pseudolabels $\boldsymbol{y^*}$ for the unlabeled samples:

$$\boldsymbol{y^*} = \operatorname{argmax}(f_\theta(T_w(\boldsymbol{u}))), \tag{1}$$

where $f_\theta(\cdot)$ is the network being trained and $T_w(\cdot)$ is a function that applies "weak" augmentations, like horizontal flipping and random cropping, to the samples.

If the confidence of the estimated label is above a user-specified threshold $\tau$, the pseudolabel is retained and used for computing the unsupervised loss term. We define $m_i$ as the indicator of which confident pseudolabels are retained: $m_i = \mathbf{1}\left(\max(f_\theta(T_w(\boldsymbol{u_i}))) > \tau\right)$. The unsupervised loss term is a consistency regularization term which encourages the network output of a strongly augmented sample to be the same as the pseudolabel estimated from the associated weakly augmented sample:

$$\ell_u = \frac{1}{\sum m_i} \sum_{i=1}^{\mu B} m_i H(\boldsymbol{y^*}, f_\theta(T_s(\boldsymbol{u_i}))), \tag{2}$$

where $B$ is the batch size, $\mu$ is FixMatch unlabeled sample ratio, $H$ is a cross-entropy loss and $T_s(\cdot)$ is a function that applies "strong" augmentations like RandAugment (Cubuk et al., 2020).

## 3.3 BACKDOOR ATTACK VULNERABILITY CONSIDERATIONS

With the consistency regularization and pseudolabeling in mind, we rethink how poisoned samples in backdoor attacks may interact differently in semi-supervised training than in supervised training.

**Augmentation-Robust Triggers** Most backdoor attacks have been analyzed in the absence of data augmentations to focus on the impact of the attack itself without introducing augmentation as a confounding factor. However, prior experiments have shown that data augmentation during training can significantly reduce the attack success rates (Li et al., 2020; Schwarzschild et al., 2021). Therefore, to understand the potential effectiveness of backdoor attacks against FixMatch, it is important to use a trigger that is robust to both the weak and strong augmentations that are crucial to its success. We

prioritize the robustness of the triggers to data augmentation over their conspicuousness in order to understand the worst case attack potential before focusing on trigger imperceptibility.

**Estimating Poisoned Labels** In backdoor attacks on supervised learning, the adversary can fix a training label for every poisoned sample and apply triggers to samples that are confusing given these training labels. This forces the network to rely on the trigger to effectively classify poisoned samples as their poisoned training labels. In attacks on the unlabeled data in semi-supervised learning, the adversary is unable to specify training labels and instead the network is responsible for estimating pseudolabels during training. This reliance on the pseudolabels of poisoned samples adds new considerations when understanding backdoor attacks. First, the adversary can try to control the expected pseudolabels through the image content itself. Second, because the pseudolabels are estimated using the current network state, the training labels assigned to poisoned samples will vary during training as the network is updated. Finally, only poisoned samples with confident network outputs will impact the network updates. We suggest that attacks against semi-supervised learning be developed and understood by considering how an adversary may vary the image content in a way that influences the expected pseudolabel outputs.

**Perturbation-Based Attack** To analyze the impact of pseudolabel behavior on attack success, we use adversarial perturbations which have been shown to successfully influence estimated network outputs. Adversarial perturbations are optimized to achieve misclassification of the images while constraining perturbation magnitude. We employ attacks that use untargeted adversarial perturbations to vary the expected pseudolabels. These attacks can vary from having no perturbations (i.e., the original training images with triggers added) to having large perturbations that significantly vary the image appearance. This is similar to the clean-label backdoor attack from Turner et al. (2019), which uses projected gradient descent (PGD) adversarial perturbations (Madry et al., 2018) to make poisoned samples more confusing to the network. However, our attack does not have training labels to constrain the network outputs. With our attack threat model in which there are limited labeled samples, we acknowledge the practical difficulty the adversary would have in obtaining enough data to fully train a network for generating adversarial attacks. We view perturbation-based attacks as a starting point for understanding how influencing pseudolabels can impact backdoor success from which future attacks can be built.

To understand how the strength of adversarial perturbations impacts the distribution of estimated network outputs, we examine the outputs from a network trained using supervised learning on CIFAR-10 training samples. Using PGD adversarial perturbations, we vary the constraint $\epsilon$ on the $\ell_\infty$ norm of the perturbation magnitude. We apply triggers and weak augmentations to the perturbed images to model the poisoned samples in semi-supervised learning. Fig. 1a shows the impact of perturbation strength on pseudolabel outputs. The blue line is the average percentage of perturbed samples with estimated network outputs that match their ground truth class and the green line is the average entropy of the distribution of class outputs for perturbed samples. As the perturbation strength increases, fewer poisoned samples are estimated to be the ground truth label and the entropy of the distribution of network outputs increases, indicating the class estimates are distributed more evenly across all class outputs. For a more granular view, Fig. 1b shows the distribution of network outputs for samples from a single class (class 0 - the airplane class) as we vary the perturbation strength. While this test is run against a fully trained network, it gives us useful insights for reasoning about the pseudolabels during semi-supervised learning. At low perturbation strength, we expect most poisoned samples have their ground truth classes as pseudolabels. At greater perturbation strength, we expect most poisoned samples will not have their ground truth classes as pseudolabels and instead their pseudolabels will be relatively evenly distributed across other classes.

## 4    ANALYSIS

We begin our analysis of the vulnerability of semi-supervised learning methods to perturbation-based attacks by considering the following experimental setup.

**Datasets** We generate attacks using the CIFAR-10 dataset (Krizhevsky et al., 2009) with 50,000 training images and 10,000 test images from 10 classes. We chose this dataset because it is a standard benchmark dataset used for studying both semi-supervised learning and data poisoning.

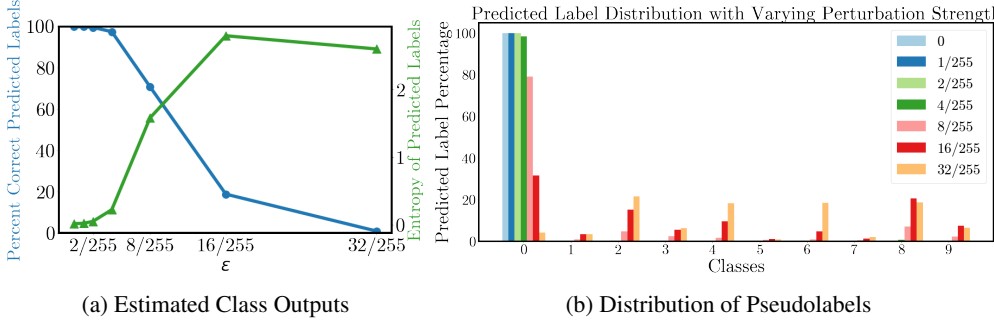

(a) Estimated Class Outputs          (b) Distribution of Pseudolabels

Figure 1: Predicted labels of perturbed samples. (a) Percentage of perturbed training samples with the ground truth class as the estimated label (blue circle line) and the entropy of the distribution of predicted labels (green triangle line) as $\epsilon$ is varied. (b) Distribution of predicted labels for samples from class 0 (airplane) as $\epsilon$ is varied.

**Semi-Supervised Learning Methods** We perform our analysis on FixMatch (Sohn et al., 2020) which achieves a classification accuracy of $94.93\%$ on CIFAR-10 with only 250 labeled samples. We largely follow the experimental details from (Sohn et al., 2020), using a WideResNet-28-2 (Zagoruyko & Komodakis, 2016) architecture, RandAugment (Cubuk et al., 2020) for strong augmentation, and horizontal flipping and cropping for weak augmentation. We experiment with 250 labeled samples. Because we are focused on analyzing the attack dynamics and define a threat model in which the adversary wants to introduce the backdoor as soon as possible during training, we limit each experiment to 100,000 training steps rather than the $2^{20}$ training steps used in the original FixMatch implementation. We found that these shorter training runs achieve relatively high classification accuracy (around $90\%$) and attacks often reach a stable state long before the end of the runs. See Appendix A for a detailed description of the FixMatch training implementation.

**Poisoning Attack** Similar to clean-label backdoor attacks, we perturb our poisoned samples using adversarially trained ResNet models (Madry et al., 2018). We define the target class of the attack as the ground truth class from which we select poisoned samples to be perturbed. Triggers are added after the images are perturbed. As discussed in Section 3.3, we begin our analysis using augmentation-robust triggers. In particular, we use the four-corner trigger, suggested in Turner et al. (2019) for its invariance to flipping and visibility under random cropping (see Fig. 5 for examples of perturbed and triggered images). This trigger is robust to strong augmentations. We define poisoning percentages with respect to the entire training set.

**Metrics** We analyze two metrics when determining the success of backdoor attacks against semi-supervised learning methods. First is the test accuracy which is the standard classification accuracy computed on the test images. Second is the attack success rate which is the percentage of non-target samples from the test set that are predicted as the target class when triggers are added to them. This indicates the strength of the backdoor in the trained network.

## 4.1 SUCCESS OF SIMPLE PERTURBATION-BASED ATTACKS

We examine the performance of simple perturbation-based backdoor attacks as we vary the constraint $\epsilon$ on the magnitude of the adversarial perturbations (see Fig. 2a). For each $\epsilon$, we run five trials, varying the target class for each run from classes 0-4, and poison $1\%$ of the entire dataset (i.e., 500 target class samples). The poisoned samples are perturbed and have the four corner trigger added. We compare the performance of the attacks against supervised learning (blue line) and semi-supervised learning (green line). Note these perturbation-based attacks against supervised learning, when the adversary sets training labels, are the same as clean-label backdoor attacks (Turner et al., 2019). The test accuracy is stable as we vary perturbation strength and the resulting accuracy with semi-supervised learning is slightly lower than the accuracy with supervised learning. This is expected because supervised learning uses all the training labels, and we are analyzing the shorter FixMatch training runs which do not reach their maximum test accuracy as detailed above.

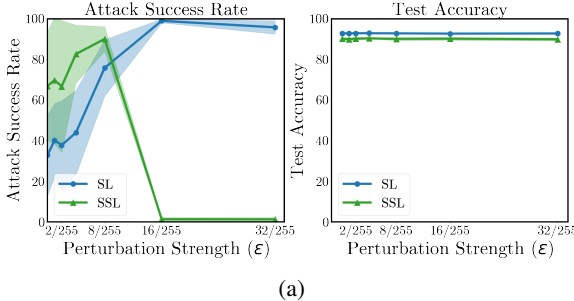 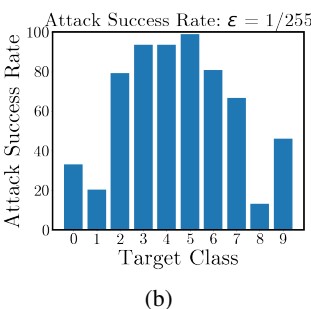

(a)                                                    (b)

Figure 2: (a) Performance of attacks against supervised learning (blue circle line) and semi-supervised learning (green triangle line) with varying $\epsilon$. (b) Attack success rate from a weak perturbation attack ($\epsilon = 1/255$) as the target class is varied.

These results show several interesting characteristics of the performance of backdoor attacks. First, the attacks against semi-supervised learning are highly successful for moderate perturbation strengths with an average attack success rate of $93.6\%$ for the attacks with $\epsilon = 8/255$ compared to an average attack success rate of $82.58\%$ for the attacks on supervised learning. Second, there is a large variation in the attack success rates for weak perturbations. Fig. 2b shows the attack success rate for each attack against semi-supervised learning with $\epsilon = 1/255$. While several attacks have very high attack success rates, the attack success rates for the attacks against classes 0, 1, and 8 are low. When comparing against supervised learning, the average attack success rate for weak perturbation attacks is high but the attacks are not consistently effective across target classes.

Turner et al. (2019) motivated the creation of their clean-label backdoor attacks against supervised learning using the fact that poisoned samples with the ground truth training label and no perturbations resulted in low attack success rates. We confirm this through the relatively low average attack success rate of $32.9\%$ from unperturbed samples ($\epsilon = 0$) against supervised learning. However, the unperturbed attack against semi-supervised learning is surprisingly effective with an average attack success rate of $73.7\%$ while also having the high variance we see with the low-perturbation attacks (see Fig. 6 for the attack success rate per target class). The final notable characteristic is the very low attack success rate for large perturbation attacks. While attack success rates against supervised learning continue to increase with larger perturbations, the attacks fail against semi-supervised learning. In Section 5 we discuss the possible reasons for this attack behavior.

## 4.2 DYNAMICS OF ATTACK SUCCESS

To understand the dynamics of backdoor attacks against semi-supervised learning, we examine the evolution of the attack success rate during training. Fig. 3a compares the attack success rates during training between supervised learning and semi-supervised learning. In supervised learning, which uses a multi-step learning rate scheduler, the attack success rate increases gradually from early in training with jumps at steps down in the learning rate. By contrast, the attack success rate during semi-supervised learning remains low for many training steps until a point in training at which it rapidly increases to a high attack success rate where it remains throughout the rest of training. This suggests that there is a tipping point at which the network forms a backdoor that strengthens rapidly. Fig. 3b shows details of the type of pseudolabels the poisoned samples have during training for attacks with weak, moderate, and strong perturbations ($\epsilon = 2/255, 8/255, 32/255$ respectively). The blue lines indicate the percentage of poisoned samples that are confidently estimated as the target class (i.e., the predicted confidence in the target class is above the threshold $\tau$). The orange lines indicate the percentage of poisoned samples that are confidently estimated as a non-target class. The green lines show the percentage of poisoned samples in which the predicted class estimates do not surpass the confidence threshold. The dashed red line is the attack success rate for reference. Of interest are the weak and moderate perturbation attacks in which the percent of poisoned samples with confident target class estimates increases steadily until a point at which nearly all poisoned samples become confident in the target class very rapidly, even if they were previously confident in another class. This suggests that as the backdoor begins to strengthen, it results in poisoned samples which were previously confusing to the network being assigned the target class as a pseudolabel.

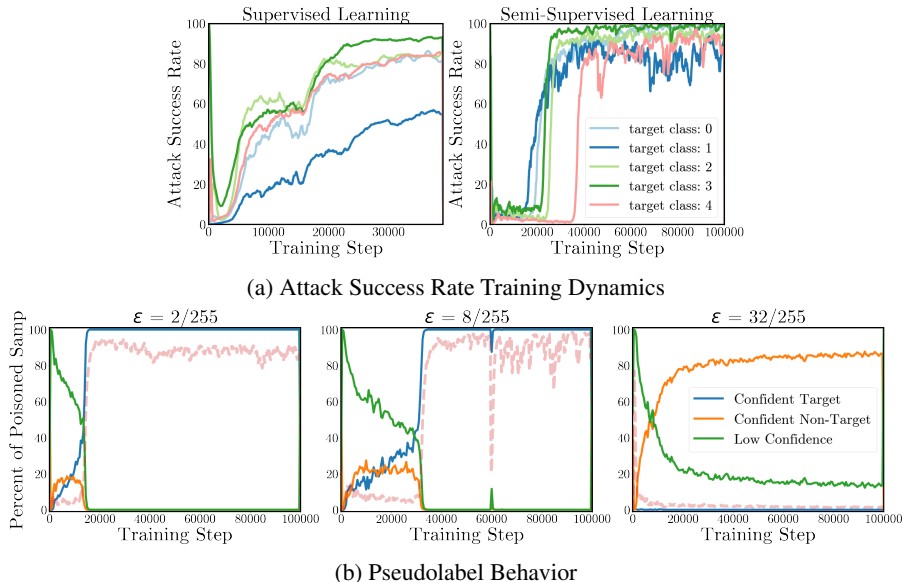

(a) Attack Success Rate Training Dynamics

(b) Pseudolabel Behavior

Figure 3: (a) Attack success rate evolution during supervised learning (left) and semi-supervised learning (right). (b) Evolution of the pseudolabel types at various $\epsilon$. The solid lines show the percentage of poisoned samples confident in the target class (blue), confident in a non-target class (orange), or not confident in any class (green). The dashed pink line shows the attack success rate.

## 5 DISCUSSION

In the previous section, we showed that simple perturbation-based attacks are very successful against semi-supervised learning models. These attacks use untargeted black-box adversarial perturbations that are generated from adversarially trained networks. In addition to the results above showing the success of attacks using weak and moderate perturbations, Appendix E shows the performance of the attacks with pretrained networks and as we vary the number of labeled samples, the percentage of poisoning, the type of trigger, and the semi-supervised learning technique. In all of these cases, we see that the moderate perturbation attacks with augmentation-robust triggers are highly effective. As we work to understand the reasons for attack success and failure on semi-supervised learning, we recognize that the perturbations influence two major factors that impact attack performance: the distribution of estimated pseudolabels and the clarity of class-specific features in the poisoned samples. We reason about the performance of the perturbation-based attacks by discussing how different perturbation strengths impact these two factors.

When the perturbations are weak or nonexistent, most poisoned samples will receive confident pseudolabels corresponding to the ground truth class label. The poisoned samples will have triggers but they will also have clear target-class-specific features that the network can use for classification, giving the network little reason to rely on the triggers. Notably, even in the weak perturbation attacks against semi-supervised learning, we are seeing high attack success rates for several target classes. However, weak perturbation attacks against some target classes, like classes 1 (automobile) and 8 (ship) shown in Fig. 2b, result in weak backdoors. This may indicate that some classes have more distinct features that the network can rely on more strongly, weakening the backdoor. The clean label backdoor attack against supervised learning encourages additional reliance on the trigger by increasing perturbation strength while fixing the training label as the ground truth class, making the samples more difficult to classify. Employing the same technique of increasing perturbation strength in the hope of improving attack performance against semi-supervised learning comes with the additional complication of the perturbations leading to different pseudolabel outputs.

We see the impact of this complication in the strong perturbation tests in which most of the samples have pseudolabels that are confident in non-target classes, as seen in the plot of $\epsilon = 32/255$ from Fig. 3b. Because the perturbations are untargeted, strong perturbations result in high entropy predicted pseudolabels distributed across many classes, as we see in Fig. 1a. Therefore, the network

sees samples containing triggers associated with several different classes, leading the network to ignore the trigger as a nuisance feature that does not aid in classification. This shows us how the dependence of semi-supervised learning on pseudolabels limits the effectiveness of perturbation-based attacks at perturbation strengths that causes too many confident non-target class pseudolabels.

Moderate perturbation strength attacks are a middle ground in which many poisoned samples will receive confident target class pseudolabels but several samples will be confidently classified as a non-target class or be confusing to the network (the orange and green lines in Fig. 3b). These confusing samples will encourage the network to rely more heavily on the triggers, strengthening the backdoor (as seen in the high attack success rate for $\epsilon = 8/255$ attacks in Fig. 2a).

This analysis suggests that consistently successful backdoor attacks require poison samples that have a pseudolabel distribution heavily concentrated on one class, which can form a weak backdoor, as well as a subset of poisoned samples that are confusing to the network, which can strengthen the backdoor. Next we discuss a generalized attack framework which moves beyond perturbation attacks to more broadly understand the necessary components for attack success and what leads to attack failure.

## 5.1 Generalized Attack Framework

Until now we have been analyzing attacks in which all the samples have the same perturbation strength. This directly links the likely pseudolabel distribution with the difficulty for a network to classify samples. As the perturbation strength increases, the samples become harder to the network to classify (encouraging a strong backdoor) but the entropy of the pseudolabel distribution also increases (encouraging the network to ignore the trigger). We decouple these two factors using a generalized attack framework which defines attacks that are composed of samples that can be used to create a weak backdoor $U_{pw}$ and samples that are used to strengthen the backdoor $U_{ps}$. The portion of samples from each of these categories is defined by $\lambda$: $N_p = \lambda|U_{pw}| + (1 - \lambda)|U_{ps}|$. Weak backdoor-creating samples should be designed to have the same pseudolabel which will be the target class. These samples can be unperturbed samples, weakly perturbed samples, or samples perturbed with strong, targeted adversarial perturbations that are expected to have confident target class pseudolabels. Backdoor-strengthening samples should be confusing to the network and they should initially have low confidence pseudolabels or confident non-target pseudolabels. These samples can be strongly perturbed samples, unperturbed samples from a class other than the target class, noisy samples, or samples interpolated between target class samples and non-target class samples.

We use this generalized attack framework to generate attacks targeting the automobile class (class 1) with results shown in Fig. 4. Fig. 4a shows attacks in which $U_{pw}$ contains unperturbed samples and $U_{ps}$ contains samples perturbed with $\epsilon = 16/255$. As $\lambda$ is decreased from 1 to 0.95, 0.4 and 0, the attack first becomes more successful with the addition of backdoor strengthening samples. However, too many backdoor strengthening samples causes the attack to fail. Fig. 4b shows attacks in which $U_{pw}$ contains perturbed samples with $\epsilon = 8/255$ and $U_{ps}$ contains samples perturbed with $\epsilon = 32/255$. At $\lambda = 0.95$, the attack becomes slightly more effective through the addition of only 25 strongly perturbed samples. However, introducing more strongly perturbed samples ($\lambda = 0.75$) leads to attack failure. These results highlight the benefits of the generalized attack framework - varying $\lambda$ can make ineffective attacks more successful, make already successful attacks more successful, and make successful attacks fail.

The large variation in attack performance due to relatively small variations in the portion of samples that are confusing to the network suggests a potential focus point for defenses against these types of attacks on semi-supervised learning. The inclusion of a small number of very confusing samples with triggers significantly reduces the impact of the attack.

While our analysis began focused on perturbation-based attacks, our results suggest that consistently successful attacks do not require perturbed samples but instead they require a large portion of poisoned samples that result in the same pseudolabel and a small portion of poisoned samples that are confusing to the network. This combination is accomplished by moderate perturbation attacks but may also be accomplished with other combinations of weak backdoor-creating samples and backdoor-strengthening samples. This suggests flexibility for adversaries which may not require them to train a robust network for generating adversarial perturbations, and it highlights considerations for users when understanding the vulnerabilities of semi-supervised learning methods.

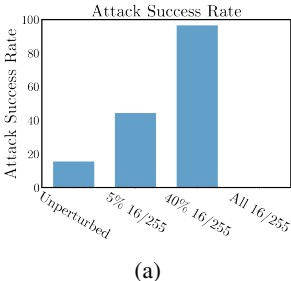
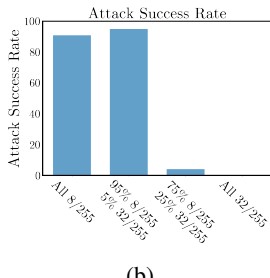

(a)                                          (b)

Figure 4: Generalized attack performance (a) $U_{pw}$ are unperturbed samples from the target class, $U_{ps}$ are perturbed samples with $\epsilon = 16/255$ for $\lambda = 1, 0.95, 0.6, 0$ (b) $U_{pw}$ perturbed samples with $\epsilon = 8/255$, $U_{ps}$ are perturbed samples with $\epsilon = 32/255$ for $\lambda = 1, 0.95, 0.75, 0$.

## 5.2 DEFENSES

We view our analysis of perturbation-based attacks against semi-supervised learning and our intro-duction of a generalized attack framework as a starting point towards understanding and defending against backdoor attacks targeting semi-supervised learning. We showed that backdoor attacks are very effective against semi-supervised learning in certain settings (i.e., with augmentation-robust triggers and moderate perturbation strength) but fail in others. This knowledge can be used to define the maximally effective attacks which can be the focus of proposed defenses.

Standard defenses that probe networks after they are trained (Liu et al., 2017; Kolouri et al., 2020; Liu et al., 2018; 2019b; Wu & Wang, 2021) should work similarly on networks trained using both supervised and semi-supervised learning because backdoor attacks have the same goal in both of those cases. Other established defenses focus on cleansing the training data by identifying poisoned samples (Chen et al., 2018; Tran et al., 2018) or reverse-engineering triggers (Wang et al., 2019; Qiao et al., 2019; Guo et al., 2019). Both activation clustering (Chen et al., 2018) and the spectral signature defense (Tran et al., 2018) identify poisoned samples by estimating clusters likely to in-clude poisoned samples using training labels which are not available in unlabeled data. Defenses that reverse-engineer triggers may more easily identify the conspicuous, augmentation-robust four corner trigger used in our analysis. This motivates future investigation into less conspicuous triggers that are also robust to significant data augmentations.

There are unique characteristics of the attacks against semi-supervised learning that suggest av-enues for future defenses. First, the labels assigned to poisoned samples in semi-supervised learn-ing vary during training. As we see in 3b, many of the poisoned samples are originally classified with pseudolabels other than the target class. This suggests that there may be an effective defense that eliminates samples that rapidly change their pseudolabel during training, limiting the backdoor strengthening samples from influencing the network. Second, we see in Figs. 2a and 4 that poi-soned samples that have confident pseudolabels associated with several classes other than the target class significantly reduce the attack success rate. This suggests further investigation into how these samples impact the attack success and how a defender may use these qualities to create a defense.

## 6 CONCLUSION

We analyzed the effectiveness of backdoor attacks on unlabeled samples in semi-supervised learn-ing when the adversary has no control over training labels. This setting requires a rethinking of attack development which focuses on the expected distribution of pseudolabels for poisoned sam-ples and the difficulty in recognizing their class-specific features. We showed that simple attacks with moderate adversarial perturbations and augmentation-robust triggers were consistently effec-tive against semi-supervised learning, and we defined a generalized attack framework which can be used to separately define weak backdoor-generating samples and backdoor-strengthening samples. This work highlights a serious vulnerability of semi-supervised learning to backdoor attacks and suggest unique characteristics of these attacks that could be used for targeting defenses in the future.

## 7 ETHICS STATEMENT

In this paper we strived to be upfront and honest about the scope of the work and its limitations so the reader has a fair understanding of what we did. We are highlighting a vulnerability of semi-supervised learning models that could be exploited by bad actors. However, we find it important to share this vulnerability with the community so practitioners can be aware of it, motivating them to check their trained models thoroughly and inspiring additional work in developing defenses against this type of attack.

## 8 REPRODUCIBILITY STATEMENT

In order to ensure reproducibility, we clearly present details of our implementations including network architectures, network parameters, and additional details that we found important for optimizing performance of our models. These details are presented in the beginning of Section 4 as well as Appendix Sections A- D. In Appendix Sections A- C we also link github repositories, code, and data that can be used for running FixMatch, generating perturbed samples, and adding triggers to poisoned samples. Finally, we provide a zip file in supplementary material including example poisoned samples for $\epsilon = 0, 1, 2, 4, 8, 16, 32/255$ that attack class 2 as well as example code showing how to incorporate those poisoned samples into a CIFAR-10 dataset for training.

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

## A   FIXMATCH TRAINING DETAILS

**Note: This section begins the supplementary appendix.**

For the FixMatch implementation, we closely follow the training set up from Sohn et al. (2020). We use a WideResNet-28-2 (Zagoruyko & Komodakis, 2016) architecture, RandAugment (Cubuk et al., 2020) for strong augmentation, and horizontal flipping and cropping for weak augmentation. We use an SGD optimizer with momentum of 0.9, a weight decay of $5 \times 10^{-4}$, and Nesterov momentum. Like Sohn et al. (2020), we use a cosine learning rate decay and quoting from them, we set the "learning rate to $\eta\cos\left(\frac{7\pi k}{16K}\right)$, where $\eta$ is the initial learning rate, $k$ is the current training step, and $K$ is the total number of training steps." We run 25,000 training epochs and each epoch runs through all the batches of the labeled data. Therefore, with 250 labeled samples, there are four steps per epoch and 100,000 steps total. We report the performance on the exponential moving average of the network parameters. We ensure an even distribution of classes in the labeled data. Additional training parameters are shown in Table 1. We found the following public github repository a good guide to implementing FixMatch: *[link to be included in final paper]*.

Table 1: Training parameters for FixMatch

| FixMatch Training Parameters |
| --- |
| batch size ($B$): 64 |
| number of epochs: 25000 |
| initial learning rate ($\eta$): 0.03 |
| total number of training steps ($K$): $2^{20}$ |
| poisoning percentage (percentage of entire dataset): 1% (500 samples) |
| number of labeled samples: 250 |
| confidence threshold ($\tau$): 0.95 |
| $\mu$: 7 |
| $\lambda_u$: 1 |

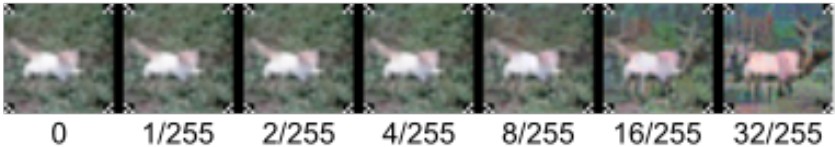

Figure 5: Poisoned images with increasing perturbation strength $\epsilon$ and the four corner trigger.

## B ADVERSARIAL PERTURBATION DETAILS

For our perturbation-based attacks we used samples that were perturbed using PGD attacks against an adversarially trained network. For $\epsilon = 8, 16, 32/255$ we used perturbed samples provided by *[details will be included in final paper]*. For $\epsilon = 1, 2, 4/255$ we used perturbed samples generated against a adversarially trained network. The adversarially trained network was a ResNet-50 using $\epsilon = 8/255$ for an $\ell_\infty$ norm. We obtained the weights for the network from *[details will be included in final paper]*.

## C POISONED SAMPLE DETAILS

We used the four corner trigger suggested in Turner et al. (2019), following the example from *[details will be included in final paper]*, for creating the attack. Fig. 5 shows an example of adversarially-perturbed poisoned images with the four corner trigger.

## D SUPERVISED LEARNING DETAILS

For supervised learning we also used a WideResNet-28-2 architecture and RandAugment data augmentation during training. We used an SGD optimizer with a momentum of 0.9 and a weight decay of $2 \times 10^{-4}$. We used a multi-step learning rate scheduler that reduced the learning rate by $\gamma = 0.1$ at epochs 40 and 60. To stay consistent with our FixMatch experiments, we report the performance on the exponential moving average of the network parameters.

Table 2: Training parameters for supervised learning

| FixMatch Training Parameters |
| --- |
| batch size: 128 |
| number of epochs: 100 |
| initial learning rate ($\eta$): 0.1 |
| poisoning percentage (percentage of entire dataset): 1% (500 samples) |

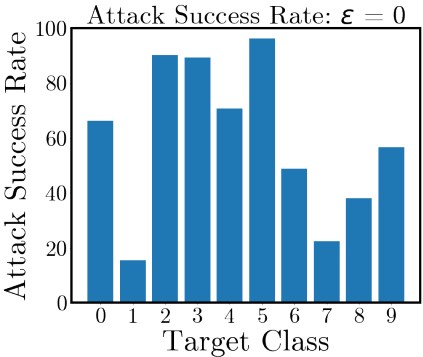

Figure 6: Attack success rate as we vary target class for attacks with unperturbed samples.

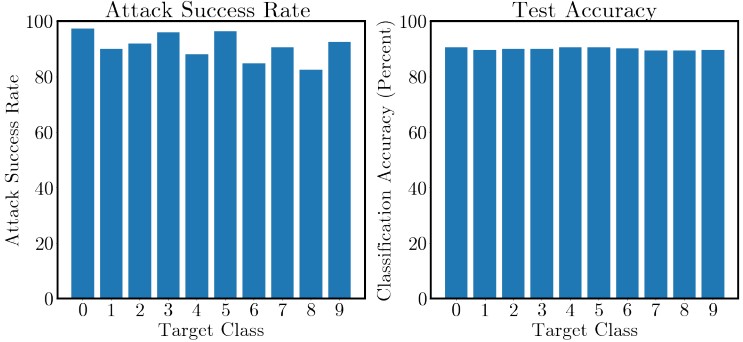

Figure 7: Attack performance as we vary target class for moderate perturbation attacks with $\epsilon = 8/255$

## E  ADDITIONAL SEMI-SUPERVISED LEARNING EXPERIMENTS

In this section we show results of additional experiments we ran to determine the attack performance varied in different settings.

**Varying Target Class**   As we showed in Fig. 2b, for attacks with weak perturbations, the attack success rate can vary significantly. The attack success rate also varies for attacks that use unperturbed samples, with some attacks achieving very high attack success rates (see Fig. 6). However, for attacks with moderate perturbation strength (like $\epsilon = 8/255$) we see fairly consistent attack success rates as we vary the target class (See Fig. 7).

**Varying Poisoning Percentage**   We examined the impact of poisoning percentage on attack performance for moderate perturbation attacks ($\epsilon = 8/255$) in Fig. 8. Note that the poisoning percentage is with respect to all 50,000 training samples in the CIFAR-10 dataset. Therefore $0.08\%$ poisoning is 40 poisoned samples and $5\%$ poisoning is 2,500 poisoned samples. The attacks fail for poisoning percentages less than $0.6\%$ after which the attack success rate increases and then plateaus.

**Varying Number of Labeled Samples**   We examine the impact of the number of labeled samples both with and without pretraining. Fig. 9a shows the performance as we vary the number of labeled samples from 250 to 4,000 and 40,000. All attacks are successful but the attack with 4,000 labeled samples has a lower attack success rate. Notably these are results for one experiment per $N_\ell$ so there may be natural variations leading to the 4,000 labeled sample run achieving the lowest attack success rate which would be evened out by averaging over multiple runs. Fig. 9b shows the attack performance as we vary the number of labeled samples and perform 20,000 training steps of pretraining with only the labeled samples prior to adding in the unlabeled samples and consistency regularization. The performance looks similar as without pretraining except with slightly lower attack success rates.

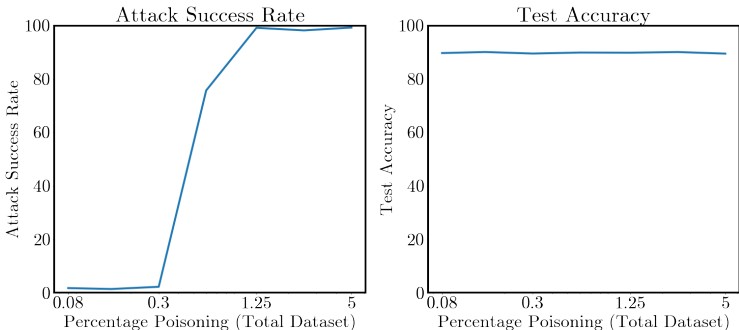

Figure 8: Attack performance as we vary poisoning percentage.

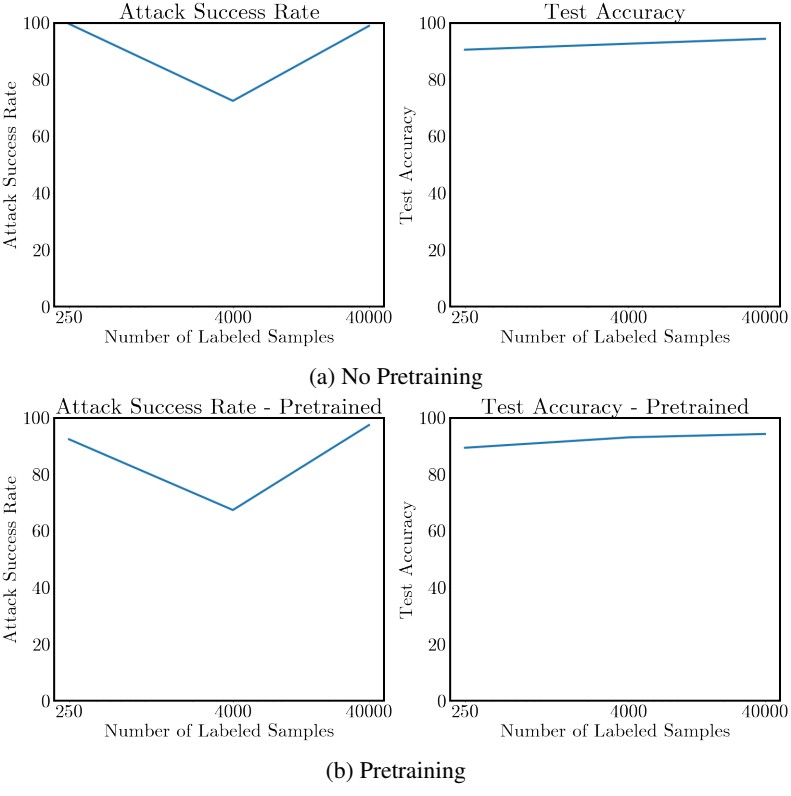

(a) No Pretraining

(b) Pretraining

Figure 9: Attack performance as we vary the number of labeled samples $N_\ell$ with and without pretraining.

Table 3: Attack performance with varying semi-supervised learning models

| Semi-Supervised Learning Technique | Attack Success Rate | Classification Accuracy |
|:---:|:---:|:---:|
| FixMatch (Sohn et al., 2020) | 99.62% | 90.52% |
| UDA (Xie et al., 2020) | 59.45% | 92.0% |

Table 4: Attack performance with varying backdoor triggers

| Trigger Type | Attack Success Rate | Classification Accuracy |
|:---:|:---:|:---:|
| Four Corner Trigger | 90.04% | 89.92% |
| $8 \times 8$ Patch Trigger | 94.50% | 89.36% |
| $4 \times 4$ Patch Trigger | 1.37% | 89.89% |

**Varying the Semi-Supervised Learning Approach**   We tested the performance of the perturbation based attack with $\epsilon = 8/255$ against the UDA semi-supervised learning technique Xie et al. (2020). This method is similar to FixMatch in its use of augmentations and consistency regularization. The main difference is that UDA computes the consistency regularization using soft network outputs rather than hard pseudolabels. Table 3 compares the performance on FixMatch and UDA on target class 0 (airplane). This preliminary experiment confirms other semi-supervised learning methods are likely to be similarly vulnerable to backdoor attacks as FixMatch.

**Vary Trigger Type**   We selected the four corner trigger which we found to be robust to strong augmentations and we used this trigger for the experiments presented in this paper. We also tested the effectiveness of single patch triggers in the bottom right of the image (See Table 4). We see that $8 \times 8$ triggers are also effective against strong augmentations but $4 \times 4$ triggers are not.

