# OpenReview forum: "Rethinking Backdoor Data Poisoning Attacks in the Context of Semi-Supervised Learning"
_ICLR.cc/2023/Conference — Submitted to ICLR 2023_

### Official Review · Reviewer_u9yW · 2022-10-23

**Confidence:** 3
**Correctness:** 3
**Technical Novelty And Significance:** 1
**Empirical Novelty And Significance:** 1
**Recommendation:** 1

**Clarity, Quality, Novelty And Reproducibility:**

The paper is reasonable clear and reproducible. In my view lacks any novelty, given the findings from Feng et al.

**Details Of Ethics Concerns:**

The paper approaches openly the sensitivity of the issue and, in my view, does a very well job, here.

**Strength And Weaknesses:**

Strength:
 - none

Weaknesses:
 - The paper, in my view, fails to claim any valid contribution. The idea to study adversarial attacks on SSL  method is not new as showed on section 2.3. There is no thorough discussion of those methods and what this paper brings in addition. Into more detail:
 * the way to carry the attack is given by prior art (Turner et al. 2019)
 * the fact that SSL methods are sensitive to attacks it has been stated before.
 * the amount of attack efficiency compared to previous works in this direction is not quantified. No comparison is carried
 * the discussion is restricted to CIFAR 10 database.
In contrast the paper by Feng et al  "Unlabeled Backdoor Poisoning in Semi-Supervised Learning" ICME 2022 discusses the problems in conjunction with two databases (CIFAR 10 and CIFAR 100), uses 2 architectures compared to 1 here. It compares different ways to attack and shows to severity of each.That paper is published into a conference that is less visible than ICLR. Expectations are higher here

The only potential contribution is that this paper tried to adapt the attack to the augmentation, yet particular and clear lessons are not drawn from this.

**Summary Of The Paper:**

The paper discusses vulnerabilities of the SSL methods to back-door attacks. The attacks are carried by poisoning unlabelled data. The SSL method of choice is FixMatch, while the analysis is carried on the CIFAR 10 dataset.

**Summary Of The Review:**

In summary, the contribution, if exists, it is not at the level required by a conference such as ICLR.

---

> ### Author Response · Authors · 2022-11-11
> **Response to Reviewer u9yW**
>
> Thank you for taking the time to review our paper. We included a general comment to all reviewers above which addresses most of the comments from your review. We have a few notes specific to your review. First, we compare the attack efficiency of perturbation based attacks between supervised learning and semi-supervised learning to highlight the unique attack behavior against semi-supervised learning. This helps us understand attack behavior and adapt both attacks and defenses aimed at semi-supervised learning in the future.
>
> In the general comment to all reviewers, we detailed the differences between our work and Feng et al’s work. They approach data poisoning in a significantly different way than we do by  training a network to make poisoned samples appear similar to target samples in the feature space and output space of the user’s network (or a proxy network). They do not have a static trigger and aim to adapt the decision boundary in a way that is reminiscent of instance-targeted attacks.

---

### Official Review · Reviewer_zodC · 2022-10-24

**Confidence:** 4
**Correctness:** 2
**Technical Novelty And Significance:** 2
**Empirical Novelty And Significance:** 2
**Recommendation:** 3

**Clarity, Quality, Novelty And Reproducibility:**

Overall, the paper is well organized and written.
In terms of the novelty, as mentioned before, backdoors attacks against semi-supervised learning algorithms are somewhat novel, but the proposed attack is just a straightforward application of the attack proposed by Turner et al. against supervised learning.
There are certain parts of the experimental settings that are not really clear, as mentioned before (generation of adversarial perturbations). In this sense, what would happen if the attacker just has a limited access to label data points? How do this impact the attack effectiveness?


**Strength And Weaknesses:**

Strengths:
+ Backdoor poisoning attacks in semi-supervised learning have not been explored in the research literature and it can pose a significant risk in some application domains. Although poisoning attacks have been proposed in this context, it is not the case for backdoors.
+ The background is well covered and the organization of the paper is good.

Weaknesses:
+ Although the context of application is novel, the proposed attack is a straightforward application of the backdoor attack proposed by Turner et al. in the context of supervised learning, which really limits the practical novelty of the paper.
+ The attack requires to train a separate model to craft the adversarial perturbations needed for the backdoor attack. From the paper, it is unclear what are the settings for doing this and this can have an impact on the threat model. For example, it does not really make sense if the authors use a supervised learning scheme to train the model and craft the adversarial examples (knowing all the labels of the complete dataset) and the defender only has access to a few labels and uses semi-supervised learning. It looks like the model for the adversary is quite strong (even if the attack is black/grey box.
+ The attack just targets one semi-supervised learning algorithm, FixMatch. It would be necessary to provide a more comprehensive evaluation against different types of semi-supervised learning algorithms. See for example the targeted attack provided by Carlini et al 2021.
+ The experimental evaluation is limited: 1) Only CIFAR dataset is used. 2) No defenses are considered (e.g. defenses against poisoning attacks in semi-supervised learning). 3) Only one semi-supervised learning method.
+ The use of untargeted attacks for generating the adversarial perturbations looks a bit odd. It seems that this is causing the low effectiveness for attacks with higher perturbations. It is unclear why the authors do not use targeted attacks instead to increase the effectiveness of the backdoor attacks as the perturbation increases.


**Summary Of The Paper:**

The authors present a backdoor data poisoning attack targeting semi-supervised learning classifiers. For this, the authors rely on a clean-label backdoor attack proposed by Turner et al. where the malicious points injected not only contain the trigger, but also an adversarial perturbation, which is necessary to achieve a high attack success rate. The experiments show the effectiveness of the attack in CIFAR-10 dataset against FixMatch, a recent method for training semi-supervised learning classifiers.

**Summary Of The Review:**

The topic addressed in the paper is interesting and novel. However, the method used for creating the attack is basically the direct application of a previous attack. There are also some aspects in the threat model / adversary capabilities that are unclear. The application of the attack by generating untargeted adversarial perturbations is also a bit strange, which explains the counterintuitive experimental results where the effectiveness of the attack is reduced when the adversarial perturbation is bigger. Finally, the experimental evaluation falls short: only one semi-supervised learning algorithm considered, no analysis of defenses, and only one dataset.
In summary, I think that the research direction is quite interesting, but the paper requires more work and a deeper analysis to be more convincing.

---

> ### Author Response · Authors · 2022-11-11
> **Response to Reviewer zodC**
>
> Thank you for your thoughtful review. We appreciate that you recognize the interesting topic we’re investigating. We included a general comment to all reviewers above and here will address specific questions that you brought up which are not covered in our above response.
> You rightly identified a limitation of this attack in practical instances of unsupervised learning when neither the attacker nor the defender will have a large amount of labeled training data. We view the adversarial perturbation-based attack as a method for probing FixMatch for vulnerabilities to backdoor attacks. In section 3.3 we discussed how backdoor attacks need to be considered in the context of their pseudolabel behavior and we used that as motivation for starting our investigation with a perturbation-based attack which we know can control expected network outputs from poisoned samples.
>
> However, we agree that extending attacks to a more realistic scenario would require moving away from the use of adversarial perturbations. This is where we see the value of the generalized attack framework in which we’ve defined a formula for attacks which include weak backdoor creating samples and backdoor strengthening samples. The generalized attack framework allows us more flexibility in defining which type of samples to use to create the backdoor and strengthen it. For instance, another way to make samples more confusing to a network is to interpolate images between target class samples and non-target class samples which does not require a trained proxy network. This suggests we could create a similarly effective attack using interpolation-based poisoned samples.
>
> We do most of our analysis on FixMatch, a popular and effective semi-supervised learning method. Given the difference in attack behavior on supervised learning and FixMatch (i.e., higher attack success rates at weak perturbation strengths and rapid increases in attack success rate during training), we are most interested in understanding how the components of FixMatch (like pseudolabeling and consistency regularization) contribute to the unique attack behavior. Therefore, we focused our investigation on a comparison between FixMatch and supervised learning rather than working across semi-supervised learning algorithms. We did confirm moderate perturbation-based attacks are successful against UDA, another semi-supervised learning method, in Appendix E.
>
> Regarding your questions about the use of untargeted adversarial perturbations: for attacks using untargeted perturbations, we set the target class to be the class from which poisoned samples are drawn, and if perturbations are unsuccessful at causing sample misclassification, that results in poisoned samples with mostly target pseudolabels. We have shown attacks with mostly target pseudolabels are often very successful. For attacks using targeted adversarial perturbations, the target class for the data poisoning attack is the class that is targeted with adversarial perturbations. This requires the targeted perturbations to be strong and well-suited to the users network in order to ensure a majority of poisoned samples are estimated as the target pseudolabel. Otherwise, the pseudolabels may be distributed across non-target classes, causing attack failure. Therefore, we chose to use untargeted adversarial perturbations because they allow us to analyze the impact of pseudolabel behavior on attack success without as many questions about the strength and transferability of the perturbations. I will note that Yan et al’s attack shows that targeted adversarial perturbations can be used to successfully poison semi-supervised learning, and samples with targeted adversarial perturbations can be used as weak backdoor creating samples in our generalized attack framework.

---

> > ### Comment · Reviewer_zodC · 2022-11-22
> > **Comments after rebuttal**
> >
> > I haver read the authors' responses and the other reviewers' comments. Thank you very much for your comments and clarifications. As stated before, I think that the topic is interesting and there's some value in the proposed approach. However, as pointed out in all the reviews there are some limitations and aspects of the paper that would require more work. Thus, I'm keeping my score, but I'd like to encourage the authors to revise and improve the paper.

---

### Official Review · Reviewer_nc1F · 2022-10-25

**Confidence:** 4
**Correctness:** 2
**Technical Novelty And Significance:** 3
**Empirical Novelty And Significance:** 3
**Recommendation:** 3

**Clarity, Quality, Novelty And Reproducibility:**

Clarity is a big issue of this paper. I've listed my concerns in the previous section.

**Strength And Weaknesses:**

**Strength**

The paper is technically sound in developing its attack techniques.

**Weakness**

Presentation is this paper's main weakness. On top of typical English language problems, this paper fails to deliver/highlight the thesis in its introduction.

Example 1. In intro, it says "In this paper we analyze the impact of backdoor data poisoning attacks on semi-supervised learning
methods to highlight a vulnerability of these methods that practitioners should be aware of when
considering the security of their models."  1) Could you, in succinct language, describe what this vulnerability is, and 2) what are "these methods" referring to? The paper primarily examines FixMatch. What are the other methods?

Example 2. In contribution, it says "We analyze the unique dynamics of data poisoning during semi-supervised training and
identify characteristics of attacks that are important for attack success." Could you explain what characteristics you have identified? The paper has a very dense experiment discussion section, in which useful insights are buried in large chunk of text. Could you highlight them in pinpoints?

Moreover, in Section 2.3, the authors have mentioned other work attacking semi-supervised learning. What is the novelty of this paper then? The related work section is not only about listing literatures but more importantly distinguishing your work from theirs.

Language-wise, there are many unnecessarily long sentences. For example on the metric used on Page 5, "Second is the attack success rate which is the percentage of non-target samples from the test set which are predicted as the target class when triggers are added to them." There are two "which" and a "when" in one running sentence with no comma... Please use short sentences if you couldn't master long ones yet. Paper writing is about clearly conveying your idea with not ambiguity.

Given the poor quality of presentation, I'm afraid that I can't tell the contribution of the paper clearly.

**Summary Of The Paper:**

This paper investigate the effectiveness of backdoor attacks against semi-supervised learning, a setting in which the learner has access to a large number of unlabeled data. The paper shows effective attacks against such a learning scheme.

**Summary Of The Review:**

Overall, this paper has some solid experiment work. However, its presentation hinders me from fully understanding the contribution and novelty. Unfortunately, I do not think it meets the ICLR standard.

---

> ### Author Response · Authors · 2022-11-11
> **Response to Reviewer nc1F**
>
> We included a general comment to all reviewers above and here will address specific questions that you brought up which are not covered in our above response.
> Regarding example 1: The vulnerability of semi-supervised learning that we highlight is its susceptibility to simple black-box backdoor attacks with adversarially perturbed samples and augmentation-robust triggers. We show how these attacks influence the distribution of pseudolabels in a manner that affects attack success. We focus our experiments on FixMatch and confirm that the attack is also successful against UDA in Appendix E.
>
> Regarding example 2: The characteristics that we have found important for attack success are an augmentation robust trigger and a combination of many poisoned samples with the same pseudolabel and a smaller set of poisoned samples that are difficult for the network to correctly classify as the target class. Importantly, if there are too many pseudolabels distributed across classes, the attacks fail. We will work to more clearly highlight these contributions in the paper.
>
> We focus on backdoor data poisoning attacks which differentiates us from Carlini, 2021 whose focus is instance-targeted attacks. In the general reviewer response above we detail the differences between our work and existing approaches that perform backdoor attacks against semi-supervised learning.

---

### Author Response · Authors · 2022-11-11
**General Response to All Reviewers**

We’d like to thank the reviewers for the time they spent reviewing our work and for their thoughtful feedback. We gave each reviewer’s comments serious consideration, and have attached a detailed response to each reviewer’s individual critique of our paper submission. Further, we have uploaded an updated PDF reflecting improvements to our submission inspired by reviewers comments (We note the most significant changes at the end of this comment to enable a more efficient review).

The purpose of our paper is to communicate to the community that there are existing backdoor attacks commonly used against supervised learning that have a very high attack success rate when applied to semi-supervised learning (SSL) with previously unpublished modifications.  In fact, following the conclusions from perturbation-based backdoor attacks against supervised learning (Turner et al, 2019), we would expect these attacks to be most successful with strong perturbations and relatively unsuccessful with weakly perturbed samples. Following this previously established understanding of effective attack settings leads to unsuccessful attacks against SSL due to lack of robustness to strong augmentations and a pseudolabel distribution that is ineffective for the creation of a backdoor. However, we show that we can ensure attack success through simple changes to existing perturbation-based attacks. This simplicity we discovered is why we feel the work needs to be urgently published, because in fact a much less sophisticated adversary would be able to construct highly successful attacks by using our proposed framework, and so the security threat to practitioners is even higher than was previously known. Since ICLR is a conference with high visibility, this awareness would spread rapidly.

An overarching theme of feedback across all reviewers was asking for clarification about the novelty of the work and why it would warrant publication in ICLR. We acknowledge that there are other more technically sophisticated methodologies in existence for crafting backdoors into models during SSL, as pointed out by the reviewers and cited in our paper (Yan el al., 2021; Feng et al., 2022). We did not intend to present an attack to be compared directly to these approaches. We wanted to highlight that we were optimizing different objectives, and feel our conclusions about highly-successful simple approaches to introducing backdoors into SSL methods complement the existing literature well.

Mindful of this, we’d like to respectfully ask the reviewers to take a fresh consideration of the paper and consider the question: “Would it be valuable for ICLR to widely disseminate the key finding that SSL approaches are widely susceptible to simple and accessible backdoor attacks?” If the answer is still no, we’d welcome your input on where to move next to disseminate this information to the community. Again, we appreciate your consideration and the time it takes to produce these reviews.

While not comparing directly to existing approaches, we would like to clarify the differences between our work and existing work. Yan et al concluded that perturbation-based clean label backdoor attacks, like what we’re using, were unsuccessful against FixMatch. This led them to develop a more complex attack which required targeted adversarial perturbations with some knowledge of the user’s network architecture and the use of a contrastive data poisoning strategy. By contrast, we identified small changes to traditional clean label backdoor attacks that enabled attack success. Additionally, our generalized attack framework allows the flexibility for targeted adversarial perturbations, like those suggested by Yan et al, to be used for weak backdoor creation.

In Feng et al, rather than using a static trigger, they train a network which encourages selected samples to have features and outputs similar to target samples in a proxy network while maintaining an appearance similar to the original samples. This is similar to instance-targeted attacks (Shafani et al, 2018) in that it inserts poisoned samples to adapt a decision boundary rather than introducing a backdoor that is associated with a specific trigger, as is the goal in most backdoor attacks including the ones we’re investigating. Therefore this work examines a different type of attack than what we’re investigating.

Updates in PDF:
- Updated Figure 4 and the associated paragraph in Section 5.1 with new results
- Updated Figure 2b to include all target classes.
- Added Figure 6 to show the attack performance with unperturbed samples as we vary the target class.
- Added details in Section 2.3 on differences between our work and Feng et al.
- Adjusted wording at the end of the Introduction and in the “Perturbation-Based Attack” paragraph of Section 3.3 to clarify that the perturbation-based attack was chosen is a way to influence pseudolabel behavior.

---

### Decision · Program_Chairs · 2023-01-20

**Decision:**

Reject

**Justification For Why Not Higher Score:**

Explained in part 1.

**Justification For Why Not Lower Score:**

N/A

**Metareview: Summary, Strengths And Weaknesses:**

The paper investigates the vulnerabilities of semi-supervised learning methods to backdoor data poisoning attacks on the unlabeled samples and shows that a simple poisoning attack using adversarially perturbed samples is highly effective. While topic is important and valuable to the community, the study is mainly limited to applying Fixmatch for semi-supervised training on Cifar10. Furthermore, the way perturbation based attacks are generated may not be very realistic for semi-supervised learning. Finally, the effectiveness of existing defenses is not studied. It would be very helpful for the community to know whether such vulnerabilities can be alleviated using existing techniques.